# Does Coenzyme Q10 Supplementation Improve Testicular Function and Spermatogenesis in Male Mice with Chronic Kidney Disease?

**DOI:** 10.3390/biology10080786

**Published:** 2021-08-17

**Authors:** Chih-Wei Tsao, Yu-Juei Hsu, Xiang-Ting Tseng, Ting-Chia Chang, Chang-Huei Tsao, Chin-Yu Liu

**Affiliations:** 1Division of Urology, Department of Surgery, Tri-Service General Hospital, National Defense Medical Center, Taipei 11490, Taiwan; weisurger@gmail.com; 2Division of Nephrology, Department of Medicine, Tri-Service General Hospital, National Defense Medical Center, Taipei 11490, Taiwan; yujuei@mail2000.com.tw; 3Department of Nutritional Science, Fu Jen Catholic University, New Taipei City 242062, Taiwan; nina00148@gmail.com (X.-T.T.); ctc5628@gmail.com (T.-C.C.); 4Department of Microbiology and Immunology, National Defense Medical Centre, Taipei 11490, Taiwan; changhuei@mail.ndmctsgh.edu.tw

**Keywords:** coenzyme Q10, chronic kidney disease, spermatogenesis, oxidative stress, mitochondria

## Abstract

**Simple Summary:**

Chronic kidney disease (CKD) is found to be linked to elevated apoptosis, oxidative stress and inflammation. Moreover, lower testosterone, poorer sperm quality and lower reproductive function have also been observed. Coenzyme Q10 (CoQ10), a widely used antioxidant involved in mitochondrial energy production, is indispensable for maintaining the efficient energy system of spermatozoa and protecting their membranes from lipid peroxidation, yet there are few studies focusing on the effects of CoQ10 on CKD-induced male infertility. This study suggests that CoQ10 increases testosterone levels and improves spermatogenesis in CKD mice.

**Abstract:**

The aim of the study was to examine the potential effects of coenzyme Q10 (CoQ10) on reproductive function in a chronic kidney disease (CKD) mouse model. Nine-week-old mice were randomly assigned to two groups: sham surgery (*n* = 18) and CKD surgery (*n* = 18). After surgery, the study groups received CoQ10 (10 mg/kg body weight dissolved in corn oil by oral gavage) or corn oil as a vehicle daily for 8 weeks. The groups that underwent 5/6 nephrectomy developed significant elevations of serum BUN and creatinine levels. The CoQ10 treatment significantly increased the serum and testicular CoQ10 levels and alleviated the poor semen quality from incomplete spermatogenesis. The testosterone concentration, in addition to the protein expression of enzymes related to testosterone biosynthesis, was also elevated, and the CKD-induced decrease in antioxidant activity in the testes was significantly ameliorated. The results suggest that CoQ10 could act against CKD-induced testicular dysfunction through improvements in the sperm function, testicular morphology, testosterone levels and related biosynthesis pathways, in addition to antioxidant activity.

## 1. Introduction

In 2017, around 69.8 million of the world’s population suffered from chronic kidney disease (CKD). [1]. According to the latest Global Burden of Disease Study, the estimated deaths from CKD numbered 1.2 million, which was almost a 42% increase from 1990 [2]. It has been reported that one of the important pathological mechanisms of CKD progression is increased oxidative stress [3,4] as observational studies of oxidative stress biomarkers have proven [5,6].

Oxidative stress also plays an influential role in the etiology of male infertility. Adequate levels of reactive oxygen species (ROS) maintain the normal function of sperm [7], but ROS overproduction depletes the levels of antioxidants, increases oxidative stress, damages the structure and quality of sperm cells, and consequently causes infertility [8,9]. In addition, lower total antioxidant activities [10] and higher levels of ROS [11] have been identified in semen from infertile men compared with that from fertile men.

Mitochondrial energy production, which is required for maintaining the spermatozoan energy system and to protect it from lipid peroxidation [12], requires coenzyme Q10 (CoQ10), a member of the ubiquinone family that has strong antioxidant properties. It is one of the most widely used antioxidants for treating idiopathic male infertility [13], yet few human or animal studies have explored the use of CoQ10 supplements to treat male infertility with CKD status. Therefore, this study investigated the effects of CoQ10 on the testicular function and spermatogenesis in male mice with CKD.

## 2. Materials and Methods

### 2.1. Experimental Design

A total of 36 male C57BL/6 mice, aged eight weeks, purchased from the National Laboratory Animal Center (Taipei, Taiwan), were housed in cages at a constant room temperature of 23 ± 3 °C, with a 12 h light/dark cycle and free access to standard food (LabDiet, 5010, St. Louis, MO, USA) and water. All the experiments were performed after obtaining approval from the Institutional Animal Care and Use Committee (IACUC; ethical code number: IACUC-19-207) at the laboratory animal facility of the National Defense Medical Center (Taipei, Taiwan). After one week of adaptation, the mice were randomly divided into the following four groups (*n* = 9 for each group): sham surgery, which received corn oil (SC); sham surgery, which received CoQ10 (SQ); CKD surgery, which received corn oil (CC); and CKD surgery, which received CoQ10 (CQ).

The CKD mouse model was established by 5/6 nephrectomy according to the method described by Gava et al. [14]. Briefly, two-thirds of the left kidney was removed in the first week, and the right kidney was completely removed in the second week. For the sham surgery, animals were operated upon in the same manner but without removal of the kidneys. CoQ10 was obtained from Sigma-Aldrich (C9538, St. Louis, MO, USA) and dissolved in corn oil to be administered at a dosage of 10 mg/kg body weight daily by gavage for 8 weeks after surgery. All the mice were then anesthetized with 1.2% Avertin solution via intraperitoneal injection and sacrificed. Blood samples were harvested from the heart and centrifuged to isolate the serum for later analysis. The reproductive organs were excised and weighed. One testis was fixed in 10% formalin (diluted from a 37% formaldehyde solution (J.T. Baker, Phillipsburg, NJ, USA)) for histological analysis, and the other was frozen in liquid nitrogen. The vas deferens were separated to flush out all the spermatozoa with a needle and syringe containing 0.5 mL of diluted phosphate-buffered saline (PBS; Bioman, Taipei, Taiwan), and the sperm supernatant fluid was collected for analysis.

### 2.2. Serum Analysis

The concentrations of blood urea nitrogen (BUN), creatinine and testosterone were analyzed by the Union Clinical Laboratory (Taipei, Taiwan). The serum CoQ10 levels were estimated using an enzyme-linked immunosorbent assay (ELISA) following the manufacturer’s protocol (MyBioSource, MBS9346778, San Diego, CA, USA).

### 2.3. Sperm Analysis

All the sperm function evaluations, including the count, motility and morphology, were performed by the same researcher. The motility was assessed using a specialized counting chamber with a special depth (Marienfeld, London, UK) to count motile and immotile sperm under a light microscope (Nikon, E400, Tokyo, Japan) at 100× magnification. At least 100 sperm were counted in each sample, and motility was recorded as the percentage of motile sperm with progressive movement. Samples were then incubated for 15 min to determine the sperm count using a cell counter (Bio-Rad, TC20, Washington, DC, USA). In addition, the samples were placed on slides and left to dry at room temperature. The slides were then fixed, stained and viewed under a light microscope (Leica, DM1000, Wetzlar, Germany). The percentage of cells having at least 100 spermatozoa of normal morphology was calculated and repeated twice.

### 2.4. Testicular Histology

The formalin-fixed testes were later examined at the Department of Pathology of Cardinal Tien Hospital (New Taipei, Taiwan). The testes were cut into 5 µm thick sections and stained with hematoxylin and eosin. The sections of the testes were visualized and images captured under a light microscope at 40×, 100× and 400× magnification (Leica, DM1000, Wetzlar, Germany) to examine the histopathological changes. The parameters, the mean seminiferous tubule diameter (MSTD) and mean testicular biopsy score (MTBS), were analyzed by the same person and calculated from at least 10 visual fields in each group.

### 2.5. Analysis of Testicular CoQ10 and Testosterone Level

Total protein from testis tissue was extracted using RIPA lysis buffer (Thermo, 89900, Waltham, MA, USA) with protease and phosphatase inhibitor cocktails and centrifuged at 14,000× *g* for 20 min. The testicular concentrations of testosterone (Cayman, 582701, Ann Arbor, MI, USA) and CoQ10 (MyBioSource, MBS9346778, San Diego, CA, USA) were measured using commercial kits according to the manufacturers’ instructions.

### 2.6. Antioxidant Capacity

The testicular antioxidant capacity consisted of the levels of superoxide dismutase (SOD), catalase (CAT) and glutathione peroxidase (GPx), which were measured using commercial kits (SOD: Cayman, 706002, Ann Arbor, MI, USA; CAT: Cayman, 707002, Ann Arbor, MI, USA; GPx: Cayman, 703102, Ann Arbor, MI, USA) according to the protocols provided by the manufacturer.

### 2.7. Western Blotting

The total protein concentrations were determined using a DC protein assay kit (Bio-Rad, 5000116, Washington, DC, USA). Twenty micrograms of protein were resolved by sodium dodecyl sulfate–polyacrylamide gel electrophoresis (SDS–PAGE) and transferred to polyvinylidene difluoride (PVDF) membranes (GE Healthcare, Freiburg, Germany). The membranes were blocked with 5% nonfat milk at room temperature for 1 h and exposed overnight at 4 °C to the primary antibodies as follows: CYP11A1 (1:1000; sc-292456), CYP17A1 (1:1000; sc-66850), 3β-HSD (1:500; sc-28206), 17β-HSD (1:250; sc-135044), StAR (1:1000; sc-25806), PGC-1α (1:250; sc-517380), NRF1 (1:1000; sc-101102), NRF2 (1:1000; sc-365949), PPARα (1:1000; sc-398394) and TFAM (1:250; sc-1666965), obtained from Santa Cruz Biotechnology (Santa Cruz, CA, USA). After rinsing in TBST buffer (Omicsbio, Taipei City, Taiwan), the membranes were incubated with secondary antibodies—HRP goat anti-mouse IgG (1:5000; Santa Cruz Biotechnology, sc-2005, Santa Cruz, CA, USA) or HRP goat anti-rabbit IgG (1:4000; Santa Cruz Biotechnology, sc-2054, Santa Cruz, CA, USA)—for 1 h at room temperature. The signals were detected using chemiluminescent imaging and analysis (MiniChemi 610, Sage Creation Science, Beijing, China) with an enhancing chemiluminescent detection reagent (Omicsbio, Taipei, Taiwan). The intensities of the protein signals were estimated using the ImageJ software, and the expression at the protein level was normalized to the expression of β-actin (1:10,000; Sigma-Aldrich, A5316, St. Louis, MO, USA).

### 2.8. Statistical Analyses

All the data are expressed as the mean ± standard deviation (SD). Statistical analyses were performed using SAS software (9.4, SAS Institute Inc., Cary, NC, USA). The mean body weight at different time points was assessed using repeated-measure analysis of variance (ANOVA). Two-way ANOVA was used to determine differences between the four groups, followed by Duncan’s new multiple-range post hoc test. A significant difference was accepted at the 95% confidence level (*p* < 0.05).

## 3. Results

### 3.1. CoQ10 Supplementation Increased Both Serum and Testicular CoQ Levels in CKD Mice

The mean serum and testicular CoQ levels of the CKD mice were remarkably reduced compared with those of the mice that received the sham surgery. In the CKD mice treated for 8 weeks with CoQ, the levels in both the serum and testes were significantly higher; however, a decreased level of serum CoQ in the SQ group was observed (Figure 1).

### 3.2. CoQ10 Supplementation Did Not Improve Renal Dysfunction in CKD Mice

The body weight, water intake, food intake and levels of BUN and creatinine were measured. Of the animals that underwent CKD surgery, the vehicle-treated group had a lower body weight and higher water intake and BUN and creatinine levels. By contrast, the CoQ10-treated group showed an increasing but nonsignificant trend in body weight, but the treatment did not restore the water intake or renal function. In addition, CoQ10 supplementation resulted in a statistically significant increase in food intake for animals in both the sham surgery and CKD surgery groups (Figure 2).

### 3.3. CoQ10 Supplementation Improved Semen Quality in CKD Mice

As expected, the mice that underwent CKD surgery had lower kidney and liver weights (Figure 3a). To observe the reproductive function in the CKD mice, the testes, epididymis and vas deferens were collected and weighed, and the semen quality was analyzed. Compared with the sham group, the CKD group had a significantly lower mean epididymis weight, and the weights of the testes and vas deferens exhibited reductions (Figure 3b), but the changes were not statistically significant. Regarding semen quality, the vehicle-treated CKD mice showed remarkable reductions in sperm motility and count and in the percentage of sperm with normal morphology. By contrast, the CoQ10-treated CKD mice maintained normal semen quality, and no significant differences were found between the SQ and CQ groups (Figure 3c).

### 3.4. CoQ10 Supplementation Preserved Testicular Morphology in CKD Mice

The four groups were comparable in terms of the histopathological changes and three other parameters: the seminiferous tubule diameter, Johnsen score, and germinal epithelium thickness. The Johnsen score was used to quantify spermatogenesis in the tubules [15]. Photomicrographs of the testes of CKD mice treated with CoQ10 showed increased numbers of spermatogenic cells in addition to more perfectly formed tubules and complete spermatogenesis. The seminiferous tubule diameters in the SQ group and the two CKD groups decreased slightly, but the diameter was not significantly different between the four groups. The mean Johnsen score was found to be lower in the untreated CKD mice compared with the sham surgery mice but higher in the CKD mice supplemented with CoQ10 than in those supplemented with corn oil. The CQ group showed a significant increase in the thickness of the seminiferous tubule epithelium compared with the CC group; in addition, a significant difference was observed between the sham surgery and CKD surgery groups treated with CoQ10 (Figure 4).

### 3.5. CoQ10 Supplementation Increased the Serum Testosterone Level and Partially Upregulated Protein Expressions of Testosterone Biosynthesis Enzymes in CKD Mice

The reduced serum testosterone levels in the CKD surgery mice that received the vehicle treatment were enhanced by treatment with CoQ10, and no significant difference between the SQ and CQ groups was observed. To further assess the potential efficacy of CoQ10 in altering this hormone, the levels of the testosterone-biosynthesis-related enzyme StAR, cytochrome P450 enzymes CYP11A1 and CYP17A1, and hydroxysteroid dehydrogenase enzymes 3β-HSD and 17β-HSD were measured. Western blot analysis revealed that StAR was reduced by 35% in the CKD mice treated with corn oil, and CoQ10 treatment significantly increased the protein expression of StAR (Figure 5 and Appendix A). Although there were no significant effects of CKD or CoQ10 treatment on the other analyzed biosynthesis enzymes, the CKD groups exhibited lower levels of cytochrome P450 enzymes.

### 3.6. CoQ10 Supplementation Enhanced Testicular Antioxidant Activity in CKD Mice

The activities of the enzymes SOD, CAT and GPx were analyzed. Although the SOD activity was not affected by CKD or CoQ10 treatment, the CKD mice receiving corn oil exhibited decreased CAT and GPx activity, reduced by 14 and 29%, respectively. In addition, compared with corn oil supplementation, CoQ10 supplementation in the CKD mice increased the CAT and GPx levels to some extent (Figure 6).

### 3.7. CoQ10 Supplementation Did Not Reverse Decrease in Protein Expression of Testicular Mitochondrial Biogenesis Markers in CKD Mice

To assess the effect of CoQ10 supplementation on mitochondrial biogenesis, the protein expression of PGC-1α, PPARα, NRF1, NRF2 and TFAM was measured. CKD surgery decreased the protein expression of TFAM; however, treatment with CoQ10 resulted in only a small increase. In contrast, CoQ10 treatment in the sham surgery group clearly elevated the PGC-1α level compared with the vehicle treatment. An increase in PGC-1α expression was observed in the SQ group compared with the CQ group; however, there was no significant difference between the CC and CQ groups (Figure 7 and Appendix A).

## 4. Discussion

The present study established an experimental mouse CKD model by the ligation of the kidneys and explored the potential protective effect of CoQ10 supplementation on CKD-induced reproductive dysfunction. The mice received 10 mg/kg/day of CoQ10, equivalent to a human dose of 80 mg/day for an average 70 kg adult based on the body surface area ratios of humans and mice [16]. Generally, CoQ10 is produced in tissues such as the liver, heart, kidney and skeletal muscle [17], and the major routes of excretion are in urine and feces [18]. Under normal conditions, endogenous CoQ10 satisfies a high proportion of the daily requirement [19], resulting in the excess being eliminated, which corresponds to the similar blood and tissue CoQ10 levels in the SC and SQ groups. Moreover, decreased CoQ10 levels have been observed in patients with CKD [20] and those with CKD undergoing dialysis [21,22], indicating an insufficient CoQ10 status during disease progression; therefore, CoQ10 treatment may be a potential therapy for CKD patients. Indeed, the total serum and testicular CoQ levels were measured by ELISA and were significantly decreased in the CKD-induced mice but enhanced after CoQ10 supplementation. However, CoQ10 is mostly present in human tissue, whereas CoQ9 is found in the tissue of rodents. To distinguish between the endogenous (mainly Q9) and exogenous (only Q10) coenzymes, high-performance liquid chromatography (HPLC) or liquid chromatography/mass spectrometry (LC–MS/MS) was applied [12,22,23].

Eight weeks after nephrectomy, the mice exhibited greatly increased serum BUN and creatinine levels, which are significant characteristics of renal dysfunction [24], and CoQ10 treatment did not alter the renal function parameters. In the studies of Fouad et al. [25] and Farrag et al. [26], reduced BUN and creatinine levels following CoQ10 supplementation were found in drug-induced acute kidney failure mouse models. Unlike in nephrectomy, the whole reserved kidney remains in drug-induced CKD, which allows CoQ10 to compensate for renal deficits. In both models, CoQ10 treatment resulted in normal histopathology for the renal corpuscles and tubules. The CKD mice were also observed to have an increased water intake and a reduced body weight, results that were in line with the findings of Gava et al. [14]. A lower body weight and higher water consumption were found in the 2-week 5/6 nephrectomy mouse model in this study. An increased urine volume was also presented, which may be related to the elevated water intake. Additionally, the CQ group showed an increase in body weight, whereas no significant differences were observed between the SQ and CC groups.

With regards to reproductive function, according to Dumanski and Ahmed [27], sex hormones, spermatogenesis and sexual dysfunction are factors affecting male fertility in CKD. These were manifested in the CKD mice, which exhibited characteristics of impaired spermatogenesis: decreased testosterone levels, damaged testicular pathology, reduced germ cell proliferation and reduced semen quality.

It was reported that CKD leads to a reduction in semen quality [28,29], and men with late-stage CKD have reduced sperm parameters compared with men with earlier-stage CKD [30]. CKD mice also exhibited diminished epididymis weights, which may have influenced sperm maturation, motility and migration from the testis to the vas deferens [31], possibly resulting in the reduced semen quality. Toxic metabolites, such as creatinine and urea nitrogen were also generated, and uremia may have contributed to testicular dysfunction, adversely affecting sperm production and fertility [32].

Following CoQ10 supplementation, however, the epididymis weight showed an increase, and improved testicular morphology and semen quality were observed. A meta-analysis published in 2013 examined three randomized controlled trials and reported that infertile men receiving exogenous CoQ10 had increased endogenous CoQ10 levels and better semen quality [33]. Later, in 2019, Alahmar [34] examined the efficacy of the forms and dosages of a 3-month CoQ10 supplementation in infertile men and found that the sperm concentration and motility were greatly changed. More significant improvement was seen in subjects taking 400 mg daily than in those taking 200 mg.

Khurana et al. [35] recruited 2419 men with stage 3–4 CKD, assessed their serum testosterone levels, and reported that 53% of the participants had levels below the normal range. Moreover, an obvious reduction in the protein expression of StAR was observed in mice with induced CKD. StAR is a critical rate-limiting enzyme in the testosterone biosynthesis pathway and assists cholesterol transport from the outer to the inner mitochondrial membrane to initiate testosterone synthesis [36]. Reddy et al. [37] demonstrated that bacterial-lipopolysaccharide-induced oxidative stress reduced StAR expression because excessive ROS led to the loss of the mitochondrial electrochemical gradient and disturbed the maturation of StAR, thereby reducing the expression of mature StAR and the transport of cholesterol and testosterone synthesis [38]. In CKD mice treated with CoQ10, the serum testosterone level and level of the related biosynthesis regulator StAR were significantly enhanced. CoQ10 and StAR are both located in the mitochondria, and it is therefore likely that the ROS-induced reduction in the expression of StAR was countered by the antioxidative function of CoQ10, with a decreased accumulation of free radicals [39]. A human study demonstrated that the serum levels of follicle-stimulating hormone and luteinizing hormone were greatly increased in infertile men after CoQ10 treatment, indicating that CoQ10 may be involved in hormone-regulated testosterone synthesis [40]. However, Banihani reviewed studies of human and animal models of CoQ10 and testosterone, and the results of all the human trials showed that CoQ10 had an insignificant influence on testosterone synthesis. However, CoQ10 appeared to have a positive effect on testosterone deficiency induced by toxic chemicals in animal studies because it could reverse free-radical-induced oxidative damage [41].

The possible mechanism of CoQ10′s action is as an antioxidant in the testicular antioxidative defense against oxidative stress because oxidative stress has been shown to adversely affect sperm function [8] and testicular function [42]. Moreover, CKD patients are exposed to elevated oxidative stress owing to an imbalanced antioxidative system, a damaged renal structure, an activated inflammation status, and increased concentrations of pro-oxidant and uremic toxins [3]. Indeed, the activities of testicular catalase and glutathione peroxidase were significantly reduced in the CKD mice and were rescued by CoQ10 treatment in the present study. These findings were similar to those of Maheshwari et al. [43], in that CoQ10 administration significantly improved the activities of antioxidative enzymes in a rat model of diabetic nephropathy. The efficacy of CoQ10 treatment was also reported in clinical trials [44,45,46]. In addition, as a marker of lipid peroxidation, the MDA levels in CKD patients were significantly decreased following CoQ10 treatment according to a systematic review published in 2018 [47].

Previous studies showed that damaged mitochondrial respiration may lead to elevated oxidative stress, apoptosis and inflammation in CKD [48,49]. The impaired mitochondrial respiration may be a result of increased oxidative stress [50,51]. Yeung et al. [22] suggested that an 8-week CoQ10 supplementation (300, 600, 1200 and 1800 mg daily, each for 2 weeks) improved mitochondrial function through reducing mitochondrion-derived oxidative stress in hemodialysis patients.

Mitochondrial dysfunction and oxidative stress are also associated with fibromyalgia, and a study by Cordero and Miguel demonstrated that, after taking 300 mg of CoQ10 per day for 40 days, patients exhibited enhanced activities of antioxidant enzymes, decreased expression of inflammatory cytokines and increased expression of PGC-1α and TFAM [52].

However, the administration of CoQ10 in this study failed to rescue the decreased expression of mitochondrial biogenesis markers in the testes caused by CKD. A possible cause of this discrepancy was the dosing in humans in this experiment, which was 80 mg/day for a man with an average weight of 70 kg, which was much lower than the dosing used in the aforementioned studies.

One limitation of our study was that the assessment of fertility was only based on testicular function and spermatogenesis and not performed using a matching test with female mice. A future study should evaluate the numbers of impregnated female mice and the mean number of embryos in a single uterus to appraise the reproductive function of male mice. Another limitation was that sperm motility was estimated by light microscopy, not computer-assisted sperm analysis (CASA). The CASA should be applied for semen analyses for increased accuracy [53].

## 5. Conclusions

The mouse model employed in this study demonstrated the adverse effects of CKD on male fertility. The results show that CoQ10 supplementation ameliorated the testicular dysfunction and poor semen quality resulting from CKD, probably owing to its antioxidant function.

## Figures and Tables

**Figure 1 biology-10-00786-f001:**
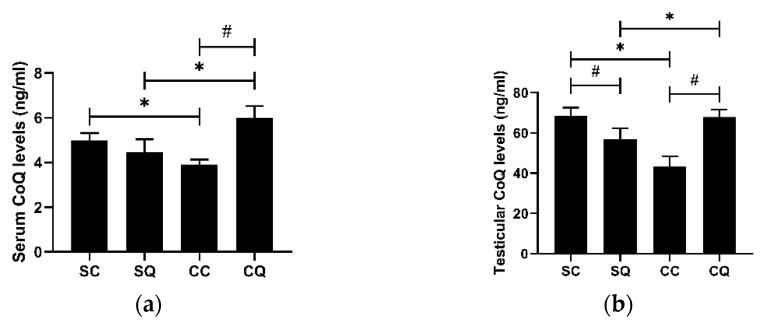
CoQ levels in (**a**) the serum and (**b**) the testes of the sham surgery and CKD surgery mice treated with vehicle (corn oil) or CoQ10. SC: sham surgery mice with vehicle; SQ: sham surgery mice with CoQ10; CC: CKD surgery mice with vehicle; CQ: CKD surgery mice with CoQ10. Data are expressed as the mean ± SD (*n* = 4–5 per group). * *p* < 0.05 denotes a significant difference between the sham surgery and CKD surgery groups (SC vs. CC and SQ vs. CQ); ^#^
*p* < 0.05 denotes a significant difference between the vehicle treatment and CoQ10 treatment groups (SC vs. SQ and CC vs. CQ). CoQ10, coenzyme Q10.

**Figure 2 biology-10-00786-f002:**
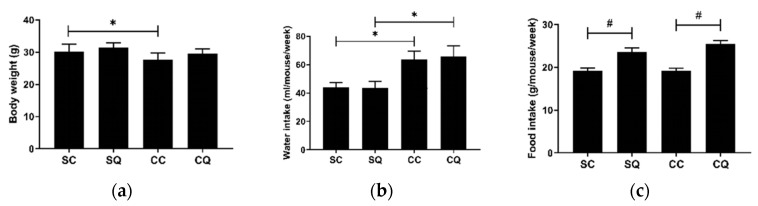
(**a**) Body weight, (**b**) water intake, (**c**) food intake, (**d**) BUN level and (**e**) creatinine level in the sham surgery and CKD surgery mice treated with vehicle (corn oil) or CoQ10. SC: sham surgery mice with vehicle; SQ: sham surgery mice with CoQ10; CC: CKD surgery mice with vehicle; CQ: CKD surgery mice with CoQ10. Data are expressed as the mean ± SD (*n* = 9 per group). * *p* < 0.05 denotes a significant difference between the sham surgery and CKD surgery groups (SC vs. CC and SQ vs. CQ); ^#^
*p* < 0.05 denotes a significant difference between the vehicle treatment and CoQ10 treatment groups (SC vs. SQ and CC vs. CQ). CoQ10, coenzyme Q10.

**Figure 3 biology-10-00786-f003:**
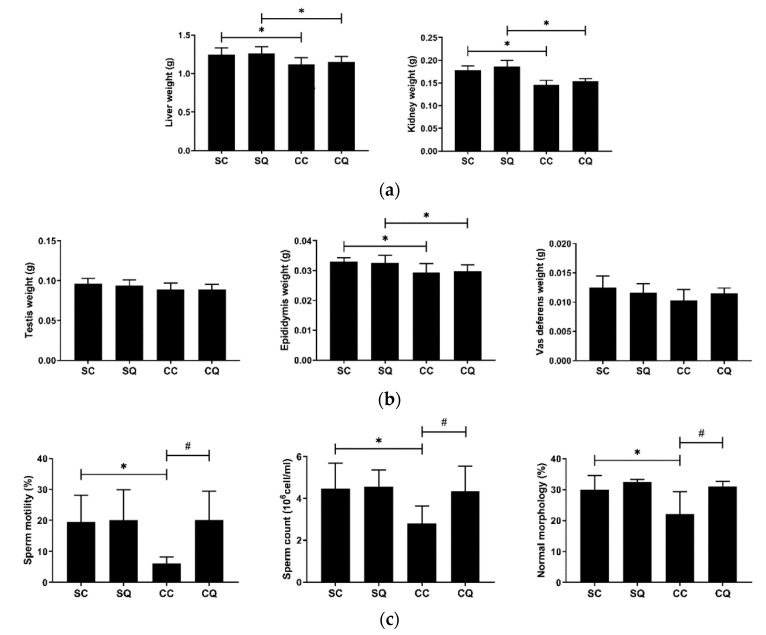
(**a**) Liver and kidney weights; (**b**) testis, epididymis and vas deferens weights; and (**c**) parameters of semen quality in the sham surgery and CKD surgery mice treated with vehicle (corn oil) or CoQ10. SC: sham surgery mice with vehicle; SQ: sham surgery mice with CoQ10; CC: CKD surgery mice with vehicle; CQ: CKD surgery mice with CoQ10. Data are expressed as the mean ± SD (*n* = 9 per group). * *p* < 0.05 denotes a significant difference between the sham surgery and CKD surgery groups (SC vs. CC and SQ vs. CQ); ^#^
*p* < 0.05 denotes a significant difference between the vehicle treatment and CoQ10 treatment groups (SC vs. SQ and CC vs. CQ). CoQ10, coenzyme Q10.

**Figure 4 biology-10-00786-f004:**
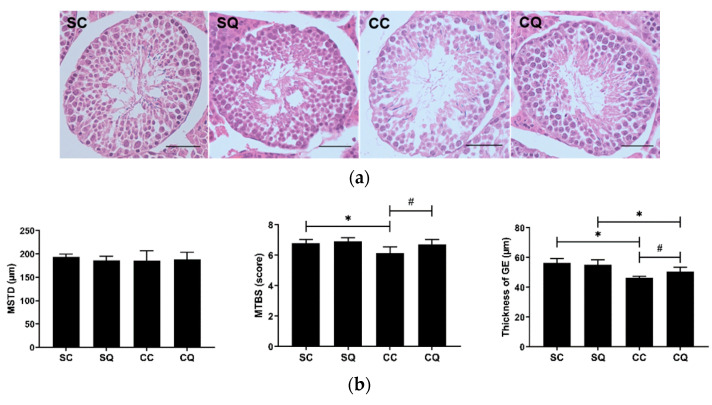
(**a**) Testicular morphology (scale bar: 50 µm), (**b**) mean seminiferous tubule diameter (MSTD), mean testicular biopsy score (MTBS) and thickness of the germinal epithelium (GE) in the sham surgery and CKD surgery mice treated with vehicle (corn oil) or CoQ10. SC: sham surgery mice with vehicle; SQ: sham surgery mice with CoQ10; CC: CKD surgery mice with vehicle; CQ: CKD surgery mice with CoQ10. Data are expressed as the mean ± SD (*n* = 5–6 per group). * *p* < 0.05 denotes a significant difference between the sham surgery and CKD surgery groups (SC vs. CC and SQ vs. CQ); ^#^
*p* < 0.05 denotes a significant difference between the vehicle treatment and CoQ10 treatment groups (SC vs. SQ and CC vs. CQ). CoQ10, coenzyme Q10.

**Figure 5 biology-10-00786-f005:**
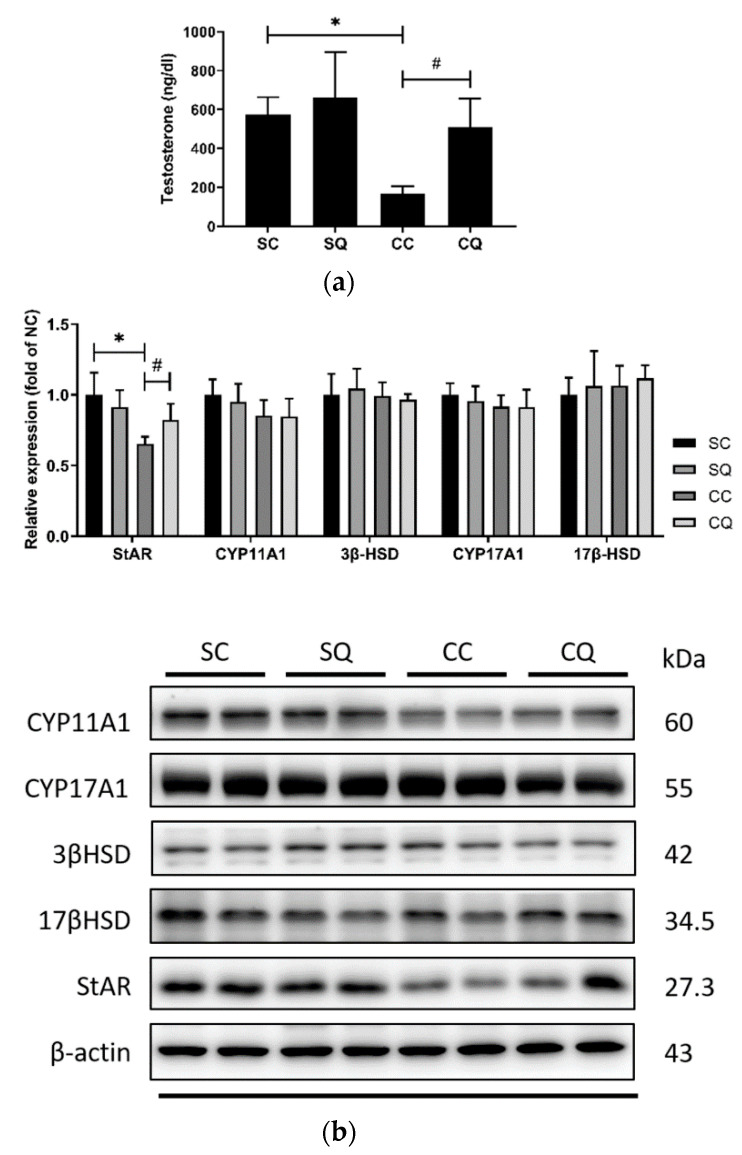
(**a**) Serum testosterone level and (**b**) protein expression of testosterone-biosynthesis enzymes (StAR, CYP11A1, 3β-HSD, CYP17A1 and 17 β-HSD) in the sham surgery and CKD surgery mice treated with vehicle (corn oil) or CoQ10. SC: sham surgery mice with vehicle; SQ: sham surgery mice with CoQ10; CC: CKD surgery mice with vehicle; CQ: CKD surgery mice with CoQ10. Data are expressed as the mean ± SD (*n* = 5–6 per group). * *p* < 0.05 denotes a significant difference between the sham surgery and CKD surgery groups (SC vs. CC and SQ vs. CQ); ^#^
*p* < 0.05 denotes a significant difference between the vehicle treatment and CoQ10 treatment groups (SC vs. SQ and CC vs. CQ). CoQ10, coenzyme Q10.

**Figure 6 biology-10-00786-f006:**
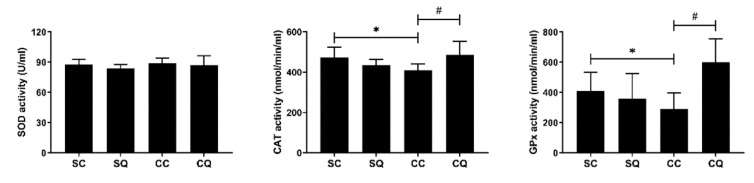
Testicular antioxidant activity in sham surgery and CKD surgery mice treated with vehicle (corn oil) or CoQ10. SC: sham surgery mice with vehicle; SQ: sham surgery mice with CoQ10; CC: CKD surgery mice with vehicle; CQ: CKD surgery mice with CoQ10. Data are expressed as the mean ± SD (*n* = 4–6 per group). * *p* < 0.05 denotes a significant difference between the sham surgery and CKD surgery groups (SC vs. CC and SQ vs. CQ); ^#^
*p* < 0.05 denotes a significant difference between the vehicle treatment and CoQ10 treatment groups (SC vs. SQ and CC vs. CQ). CoQ10, coenzyme Q10. SOD, superoxide dismutase; CAT, catalase; GPx, glutathione peroxidase.

**Figure 7 biology-10-00786-f007:**
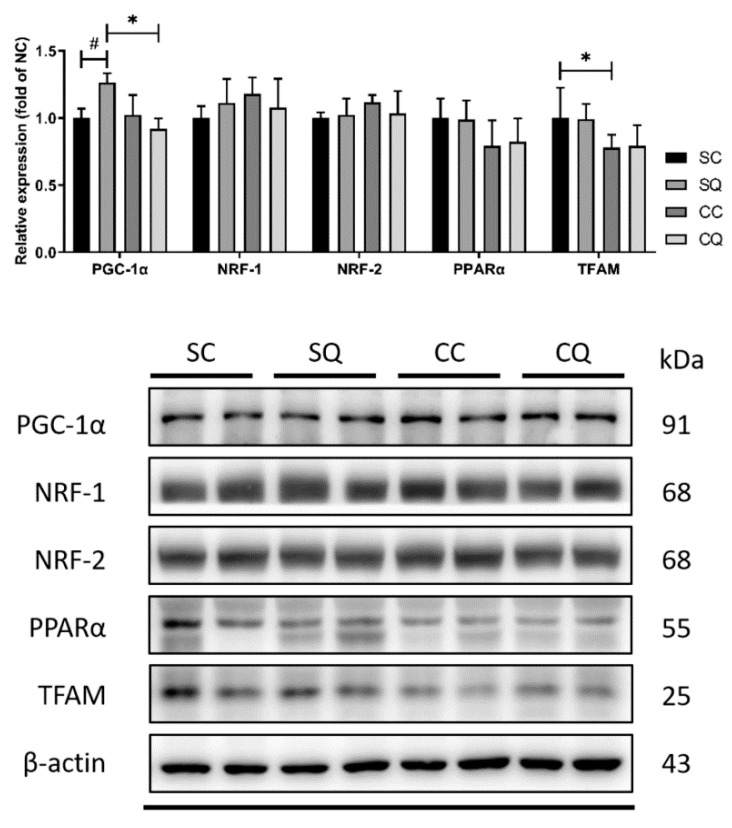
Protein expression of testicular mitochondrial biogenesis markers in sham surgery and CKD surgery mice treated with vehicle (corn oil) or CoQ10. SC: sham surgery mice with vehicle; SQ: sham surgery mice with CoQ10; CC: CKD surgery mice with vehicle; CQ: CKD surgery mice with CoQ10. Data are expressed as the mean ± SD (*n* = 4–6 per group). * *p* < 0.05 denotes a significant difference between the sham surgery and CKD surgery groups (SC vs. CC and SQ vs. CQ); ^#^
*p* < 0.05 denotes a significant difference between the vehicle treatment and CoQ10 treatment groups (SC vs. SQ and CC vs. CQ). CoQ10, coenzyme Q10.

## Data Availability

The datasets used and/or analyzed during the current study are available from the correspond-ing author on reasonable request.

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
