# Peer review of "Does Coenzyme Q10 Supplementation Improve Testicular Function and Spermatogenesis in Male Mice with Chronic Kidney Disease?"

_biology, 2021, doi:10.3390/biology10080786_

Round 1
Reviewer 1 Report
Interesting paper by Tsao et al explioring aspects linking CKD and infertility that are of interest in relation to their association with oxidative stress and therefore justify intervention with COQ10 that has an established role in fertility.
I have some doubts on some aspects related to CoQ quantification and some missing elements in the paper that should be addressed. It is true that CoQ10 has been used in several murine studies, however the major form of CoQ in rodents is CoQ9, this aspects should at least be mentioned somewhere in the introduction since all the discussion focusses on CoQ10.
This leaves me also some doubts on Coenzyme Q quantification. The method used is an ELISA assay, which is very unusual and definitely not the best quantitative method for CoQ10. I did look into the details of the kit used and the producer do not make any comment, or I could not find it, in relation to the antibody used and more specifically what it targets, if it can discriminate among Q9 and Q10, since this would be a critical aspects in the discrimination between endogenous (which is mainly Q9) and exogenous (only Q10).
I believe that these aspects deserve further insight into the methodologies and appropriate comments in the discussion.
Moreover I have some difficulties in understanding the last phrase line 295-298. In my understanding the authors comment that the effects are reported to what present in the literature because the dosage used in rodents would be equivalent in a human to 80/mg Kg and therefore they conclude lower that inthe human clinical trial. However in a 70 kg human beign 80mg/kg would result in a massive 5.6 grams dose that has never being used in any human study . Probably I'm getting something wrong and hence this last sentence needs to be rephrased because is not very clear.
Author Response
Assistant Editor
Biology
Dear Professor Annie Ji:
Please find enclosed our revised original paper entitled “Does the Testicular Function and Spermatogenesis of Male Mice with Chronic Kidney Disease Benefit from Coenzyme Q10 Supplementation?”. We appreciated the comments and suggestions provided to further improve our manuscript.
Sincerely yours
Corresponding author: Chin-Yu Liu, Ph.D.
Nutritional Science, Fu Jen Catholic University, New Taipei City, Taiwan
No.510, Zhongzheng Rd., Xinzhuang Dist., New Taipei City 24205, Taiwan.
Telephone: +886-2-29053610
Fax: +886-2-29021215
e-mail: nf351.lab@gmail.com
Comments of Reviewer 1 to Author:
Interesting paper by Tsao et al exploring aspects linking CKD and infertility that are of interest in relation to their association with oxidative stress and therefore justify intervention with COQ10 that has an established role in fertility.
I have some doubts on some aspects related to CoQ quantification and some missing elements in the paper that should be addressed. It is true that CoQ10 has been used in several murine studies, however the major form of CoQ in rodents is CoQ9, this aspects should at least be mentioned somewhere in the introduction since all the discussion focusses on CoQ10.
This leaves me also some doubts on Coenzyme Q quantification. The method used is an ELISA assay, which is very unusual and definitely not the best quantitative method for CoQ10. I did look into the details of the kit used and the producer do not make any comment, or I could not find it, in relation to the antibody used and more specifically what it targets, if it can discriminate among Q9 and Q10, since this would be a critical aspects in the discrimination between endogenous (which is mainly Q9) and exogenous (only Q10). I believe that these aspects deserve further insight into the methodologies and appropriate comments in the discussion.
Our response:
We thank the reviewer for the professional suggestion. Before performing the experiment, we did check the protocol provided by Mybiosource.com (MyBioSource, MBS9346778, California, USA) (https://www.mybiosource.com/mouse-elisa-kits/coenzyme-q10-coq10/9346778 ). The species reactivity of is for mouse and the kit used CoQ10 antibody-CoQ10 antigen interactions (immunosorbency) and an HRP colorimetric detection system to detect CoQ10 antigen targets in samples and to detect native, not recombinant, CoQ10. In our study, the serum and testis total CoQ levels were measured by ELISA and significantly decreased in the CKD-induced mice but enhanced after CoQ10 supplementation. Indeed, Q10 is the most prevalent form in humans and Q9 is the primary form found in rats, mice, and guinea pigs. If we need to display the different forms of these coenzymes, HPLC or LC-MS/MS should be the better way to identify these two [1-3]. Therefore, we correct the figure 1 and have some comments in the introduction and discussion.
- Aberg, F.; Appelkvist, E.L.; Dallner, G.; Ernster, L. Distribution and redox state of ubiquinones in rat and human tissues. Arch Biochem Biophys 1992, 295, 230-234, doi:10.1016/0003-9861(92)90511-t.
- Lekli, I.; Das, S.; Das, S.; Mukherjee, S.; Bak, I.; Juhasz, B.; Bagchi, D.; Trimurtulu, G.; Krishnaraju, A.V.; Sengupta, K., et al. Coenzyme Q9 provides cardioprotection after converting into coenzyme Q10. J Agric Food Chem 2008, 56, 5331-5337, doi:10.1021/jf800035f.
- Yeung, C.K.; Billings, F.T.t.; Claessens, A.J.; Roshanravan, B.; Linke, L.; Sundell, M.B.; Ahmad, S.; Shao, B.; Shen, D.D.; Ikizler, T.A., et al. Coenzyme Q10 dose-escalation study in hemodialysis patients: safety, tolerability, and effect on oxidative stress. BMC nephrology 2015, 16, 183, doi:10.1186/s12882-015-0178-2.
Moreover I have some difficulties in understanding the last phrase line 295-298. In my understanding the authors comment that the effects are reported to what present in the literature because the dosage used in rodents would be equivalent in a human to 80/mg Kg and therefore they conclude lower that in the human clinical trial. However in a 70 kg human beign 80mg/kg would result in a massive 5.6 grams dose that has never being used in any human study . Probably I'm getting something wrong and hence this last sentence needs to be rephrased because is not very clear.
Our response:
We thank the reviewer for the academic suggestion. We apologize for making a typo of the unit and correct it as 80 “mg/day” not “mg/kg/day”. The mice received 10 mg/kg/day CoQ10, equivalent to a human dose of 80 mg/day for an average 70-kg adult based on the body surface area ratio of humans and mice.
Reviewer 2 Report
Tsao et al. investigated the effect of CoQ10 supplementation on the recovery of reproductive function in the nephrectomized male mice as a chronic kidney disease model. Administration of CoQ10 at a dosage of 10 mg/kg body weight/day for 8 weeks in the nephrectomized mice improved the number of sperm, sperm motility, and sperm morphology (Figure 3c) as well as preserving normal testicular morphology (Figure 4), but not improved renal dysfunction (in Figure 2d and e). CoQ10 supplementation also improved the serum testosterone levels with the expression of StAR protein in the nephrectomized male mice (Figure 5). The mechanism of improving reproductive function by CoQ10 supplementation was associated with the recovery of testicular antioxidant activity but not with the recovery of mitochondria function.
This beneficial function of the CoQ10 may be interesting, but there are unclear things and problems in the experimental designs.
- The authors did not determine the male mouse fertility directly. Therefore, a mating test with female mice should be performed. Moreover, the numbers of impregnated female mice and the mean number of embryos in a single females uterus should be determined, as described in a paper by Ghanayem et al. (PMC: 19696015).
- Is it suitable to use phosphate-buffered saline (PBS) when isolating sperm? To keep the sperm function in vitro, other media such as M2 media is suitable than PBS. (PMID: 15036976). Ghanayem et al. also used M2 media when isolating and determining the sperm quality. Besides, short-time storage in PBS might damage the sperm function.
- Quantitative parameters of sperm motility should be determined using a computer-assisted sperm analysis (CASA) like Ghanayem et al. Observation under light microscopy is not suitable for evaluating the sperm quality objectively.
- In statistical analysis, what method did the authors use for the post-hoc test? Bonferroni test, LSD test, Tukey's test, or Dunnet test? Please describe it precisely.
- Please describe the method determining the antioxidant enzymes’ activities in the materials and the methods section.
- The endogeneous CoQ in mice is CoQ9, not CoQ10. Therefore, The description of CoQ10 in Figure 1 should be CoQ or CoQ9 + CoQ10. The reviewer recommends the HPLC method or LC/MS/MS method to determine both CoQ9 and CoQ10 levels at a time.
- Why did the authors use 5/6 nephrectomized mice as a model of CKD in this study? Did the authors expect the improvement of reproductive function without recovery of renal function by exogenous CoQ10? To assess the reduction of reactive oxygen species by CoQ10 supplementation, recovery of renal function is the most important (refs. 50 and 51). In the condition in which the renal function cannot be improved, exogenous CoQ10 by supplementation just netrilize the uremic toxin-derived ROS.
Author Response
Assistant Editor
Biology
Dear Professor Annie Ji:
Please find enclosed our revised original paper entitled “Does the Testicular Function and Spermatogenesis of Male Mice with Chronic Kidney Disease Benefit from Coenzyme Q10 Supplementation?”. We appreciated the comments and suggestions provided to further improve our manuscript.
Sincerely yours
Corresponding author: Chin-Yu Liu, Ph.D.
Nutritional Science, Fu Jen Catholic University, New Taipei City, Taiwan
No.510, Zhongzheng Rd., Xinzhuang Dist., New Taipei City 24205, Taiwan.
Telephone: +886-2-29053610
Fax: +886-2-29021215
e-mail: nf351.lab@gmail.com
Comments of Reviewer 2 to Author:
Tsao et al. investigated the effect of CoQ10 supplementation on the recovery of reproductive function in the nephrectomized male mice as a chronic kidney disease model. Administration of CoQ10 at a dosage of 10 mg/kg body weight/day for 8 weeks in the nephrectomized mice improved the number of sperm, sperm motility, and sperm morphology (Figure 3c) as well as preserving normal testicular morphology (Figure 4), but not improved renal dysfunction (in Figure 2d and e). CoQ10 supplementation also improved the serum testosterone levels with the expression of StAR protein in the nephrectomized male mice (Figure 5). The mechanism of improving reproductive function by CoQ10 supplementation was associated with the recovery of testicular antioxidant activity but not with the recovery of mitochondria function. This beneficial function of the CoQ10 may be interesting, but there are unclear things and problems in the experimental designs.
The authors did not determine the male mouse fertility directly. Therefore, a mating test with female mice should be performed. Moreover, the numbers of impregnated female mice and the mean number of embryos in a single females uterus should be determined, as described in a paper by Ghanayem et al. (PMC: 19696015).
Our response:
We thank the reviewer for the professional comment. The study design indeed lack of direct determine of fertility and should be considered as the limit of study. The mating test should be a better wat to reveal the protective effect of CoQ10 on CKD mice, but we still also tried integrating the indirect determines of fertility through hormone levels, semen quality and testicular morphology. Therefore, we revised the title as “Does the Testicular Function and Spermatogenesis of Male Mice with Chronic Kidney Disease Benefit from Coenzyme Q10 Supplementation?” to fit the content of study design. We also added the description about the limitation of our study in the part of Discussion.
Is it suitable to use phosphate-buffered saline (PBS) when isolating sperm? To keep the sperm function in vitro, other media such as M2 media is suitable than PBS. (PMID: 15036976). Ghanayem et al. also used M2 media when isolating and determining the sperm quality. Besides, short-time storage in PBS might damage the sperm function.
Our response:
We referenced the articles which take phosphate-buffered saline as means of isolating sperm as well. Also, dilute PBS was used to less damage sperm function, and sufficient numbers of motile sperm were observed under microscope. Indeed, more studies recently used suitable medium such as M2 buffer, we are going to try the medium from PBS to M2 buffer in further studies. It is a very important suggestion to our Lab to improve our study and thank the reviewer for the comment.
- Odet F, Pan W, Bell TA, et al. The Founder Strains of the Collaborative Cross Express a Complex Combination of Advantageous and Deleterious Traits for Male Reproduction [published correction appears in G3 (Bethesda). 2016 Mar;6(3):767]. G3 (Bethesda). 2015;5(12):2671-2683. Published 2015 Oct 19. doi:10.1534/g3.115.020172
- GüleÅŸ Ö, Kum Åž, Yıldız M, BoyacıoÄŸlu M, Ahmad E, Naseer Z, Eren Ü. Protective effect of coenzyme Q10 against bisphenol-A-induced toxicity in the rat testes. Toxicol Ind Health. 2019 Jul;35(7):466-481. doi: 10.1177/0748233719862475. PMID: 31364507.
- L Shi, T Zhou, Q Huang, SY Zhang, W Li, L Zhang, RA Hess, GJ Pazour, Z Zhang. Intraflagellar transport protein 74 is essential for mouse spermatogenesis and male fertility by regulating axonemal microtubule assembly in mice. bioRxiv 457804; doi: https://doi.org/10.1101/457804
Quantitative parameters of sperm motility should be determined using a computer-assisted sperm analysis (CASA) like Ghanayem et al. Observation under light microscopy is not suitable for evaluating the sperm quality objectively.
Our response:
We thank the reviewer for the comment. To determine the sperm motility, we used a specialized counting chamber with special depth (Marienfeld, London, UK) to count the motile and immotile sperm. Additionally, the researcher has been trained to be familiar with whole process of the semen analyses. However, we agreed that the computer-assisted sperm analysis (CASA) is more appropriate for sperm motility to decrease experimental error due to man-made observation, herein we added the description about the limitation of our study in the part of Discussion:
“One limitation of our study was noticed that the assessment of fertility being not evaluated by the matching test with female mice, only judged by the testicular function and spermatogenesis stage. Future study should be evaluated the numbers of impregnated female mice and the mean number of embryos in a single females uterus to appraise the reproductive function of male mice. Another limitation was found that the judgment of sperm motility being not estimated by the computer-assisted sperm analysis (CASA), otherwise assessed by the light microscopy. The CASA should be applied for semen analyses with more accuracy in further study [55].”
In statistical analysis, what method did the authors use for the post-hoc test? Bonferroni test, LSD test, Tukey's test, or Dunnet test? Please describe it precisely.
Our response:
We thank the reviewer for the comment. The Duncan’s new multiple range test for the post-hoc test was applied and we have added the describing in the manuscript:
“2.7. Statistical analyses
All data were expressed as the mean ± standard deviation (SD). Statistical analyses were performed using SAS software (9.4, SAS Institute Inc., North Carolina, USA). The mean body weight at different time points was assessed using repeated-measure analysis of variance (ANOVA). Two-way ANOVA was used to determine differences among four groups, and followed by Duncan’s new multiple range post hoc test. A significant difference was accepted at the 95% confidence level (p< 0.05).”
Please describe the method determining the antioxidant enzymes’ activities in the materials and the methods section.
Our response:
We thank the reviewer for the comment. We have added the related describing in the section of method in the manuscript:
“2.6. Antioxidant capacity
Testicular antioxidant capacity includes superoxide dismutase (SOD), catalase (CAT) and glutathione peroxidase (GPx) were measured using commercial kits (SOD: Cayman, 706002, Michigan, USA; CAT: Cayman, 707002, Michigan, USA; GPx: Cayman, 703102, Michigan, USA) and according to the protocols provided by the manufacturer.”
The endogeneous CoQ in mice is CoQ9, not CoQ10. Therefore, The description of CoQ10 in Figure 1 should be CoQ or CoQ9 + CoQ10. The reviewer recommends the HPLC method or LC/MS/MS method to determine both CoQ9 and CoQ10 levels at a time.
Our response:
We thank the reviewer for the comment. In our study, the serum and testis total CoQ levels were measured by ELISA and significantly decreased in the CKD-induced mice but enhanced after CoQ10 supplementation. Indeed, Q10 is the most prevalent form in humans and Q9 is the primary form found in rats, mice, and guinea pigs. If we need to display the different forms of these coenzymes, HPLC, Mass Spectroscopy or LC-MS/MS should be the better way to identify these two [1-3]. Therefore, we correct the figure 1 and have some comments in the introduction and discussion.
- Aberg, F.; Appelkvist, E.L.; Dallner, G.; Ernster, L. Distribution and redox state of ubiquinones in rat and human tissues. Arch Biochem Biophys 1992, 295, 230-234, doi:10.1016/0003-9861(92)90511-t.
- Lekli, I.; Das, S.; Das, S.; Mukherjee, S.; Bak, I.; Juhasz, B.; Bagchi, D.; Trimurtulu, G.; Krishnaraju, A.V.; Sengupta, K., et al. Coenzyme Q9 provides cardioprotection after converting into coenzyme Q10. J Agric Food Chem 2008, 56, 5331-5337, doi:10.1021/jf800035f.
- Yeung, C.K.; Billings, F.T.t.; Claessens, A.J.; Roshanravan, B.; Linke, L.; Sundell, M.B.; Ahmad, S.; Shao, B.; Shen, D.D.; Ikizler, T.A., et al. Coenzyme Q10 dose-escalation study in hemodialysis patients: safety, tolerability, and effect on oxidative stress. BMC nephrology 2015, 16, 183, doi:10.1186/s12882-015-0178-2.
Why did the authors use 5/6 nephrectomized mice as a model of CKD in this study? Did the authors expect the improvement of reproductive function without recovery of renal function by exogenous CoQ10? To assess the reduction of reactive oxygen species by CoQ10 supplementation, recovery of renal function is the most important (refs. 50 and 51). In the condition in which the renal function cannot be improved, exogenous CoQ10 by supplementation just netrilize the uremic toxin-derived ROS.
Our response:
We thank the reviewer for the comment. We did assess several CKD models to address specific hypotheses in our research. The reasons that we use 5/6 nephrectomized mice as a model of CKD in this study are:
- to investigate the morbidity (testicular dysfunction and poor semen quality) direct associated with CKD, and
- to avoid other parameters (such as drug toxicity, immune interference, virus, high blood sugar…) involved in sperm damaging.
The 5/6 nephrectomy is a physical and non-immune-induced model with a known etiology and used to mimic the progressive kidney disease when renal mass loss in humans. In our study, CoQ10 treatment did not restore the renal function but improved the number of sperm, sperm motility, and sperm morphology as well as preserving normal testicular morphology. The possible mechanism of CoQ10 action is its role as an antioxidant in the testicular antioxidative defense against CKD-induced oxidative stress, because oxidative stress has been demonstrated to have negative impacts on sperm function and testicular function.
- Yang HC, Zuo Y, Fogo AB. Models of chronic kidney disease. Drug Discov Today Dis Models. 2010;7:13-19. doi: 10.1016/j.ddmod.2010.08.002.
- Lim BJ, Yang HC, Fogo AB. Animal models of regression/progression of kidney disease. Drug Discov Today Dis Models. 2014;11:45-51. doi: 10.1016/j.ddmod.2014.06.003.
Round 2
Reviewer 2 Report
The authors have made their points, which the reviewer understands and sympathizes with, that resources have restricted the experimental design. However, his views are the same, that the experimental design has endangered the accurate interpretation of the actual outcomes, which casts serious doubt over the validity of the conclusions drawn from the manuscript.
The question is whether the reviewer should accept publishing the results simply because it was impossible to conduct studies in the way they should. Although he may acknowledge these difficulties and challenges, he does not feel it valid to publish data. It will impact the quality of science in peer-reviewed journals and negates the peer review process if it becomes acceptable not to follow the principles of scientific integrity in experimental design. The reviewer must therefore stand by his original decision and recommends that the manuscript is rejected. He feels it is up to the editors to decide whether the authors' defense of the manuscript warrants further consideration.
Author Response
Assistant Editor
Biology
Dear Professor Annie Ji:
Please find enclosed our revised original paper entitled “Does Coenzyme Q10 Improve Testicular Function and Spermatogenesis of Male Mice with Chronic Kidney Disease?”. We appreciated the comments and suggestions provided to further improve our manuscript.
Sincerely yours
Corresponding author: Chin-Yu Liu, Ph.D.
Nutritional Science, Fu Jen Catholic University, New Taipei City, Taiwan
No.510, Zhongzheng Rd., Xinzhuang Dist., New Taipei City 24205, Taiwan.
Telephone: +886-2-29053610
Fax: +886-2-29021215
e-mail: nf351.lab@gmail.com
Comments of Reviewer 2 to Author:
The authors have made their points, which the reviewer understands and sympathizes with, that resources have restricted the experimental design. However, his views are the same, that the experimental design has endangered the accurate interpretation of the actual outcomes, which casts serious doubt over the validity of the conclusions drawn from the manuscript.
The question is whether the reviewer should accept publishing the results simply because it was impossible to conduct studies in the way they should. Although he may acknowledge these difficulties and challenges, he does not feel it valid to publish data. It will impact the quality of science in peer-reviewed journals and negates the peer review process if it becomes acceptable not to follow the principles of scientific integrity in experimental design. The reviewer must therefore stand by his original decision and recommends that the manuscript is rejected. He feels it is up to the editors to decide whether the authors' defense of the manuscript warrants further consideration.
Our response:
We thank the reviewer for the unique comments. The global disease burden of chronic kidney disease (CKD) is growing rapidly, and dietary control and nutrition intervention remain the major guidelines for improving both the CKD progression and quality of life particularly in young adults. Many men in CKD exhibit subfertility or infertility due to several factors including chronic inflammation, hypertension, hyperglycemia, hypogonadism, oligospermia, erectile dysfunction and so on. It is important to address the patient’s fertility concerns thus the coauthors and I studied hard to investigate the researches between the nutrient supplementation and testicular function in individual clinical disease state:
- Chih-Wei Tsao, Yu-Juei Hsu, Ting-Chia Chang, Sheng-Tang Wu, Tai-Lung Cha, Chin-Yu Liu. A High Phosphorus Diet Impairs Testicular Function and Spermatogenesis in Male Mice with Chronic Kidney Disease. Nutrients. 2020 Aug 28;12(9):2624. doi: 10.3390/nu12092624
- Chin-Yu Liu, Ting-Chia Chang, Shyh-Hsiang Lin, Sheng-Tang Wu, Tai-Lung Cha, Chih-Wei Tsao. Metformin Ameliorates Testicular Function and Spermatogenesis in Male Mice with High-Fat and High-Cholesterol Diet-Induced Obesity. Nutrients. 2020 Jun 29;12(7):1932. doi: 10.3390/nu12071932.
The previous our reference 1 study was also the CKD mouse model developed by performing a 5/6 nephrectomy, to evaluate the testicular function and spermatogenesis with or without a high phosphorus diet. The reasons that we use 5/6 nephrectomized mice as a model of CKD in studies are: (1) to investigate the morbidity (testicular dysfunction and poor semen quality) direct associated with CKD, and, (2) to avoid other parameters (such as drug toxicity, immune interference, virus, high blood sugar…) involved in sperm damaging. In this study, if the CKD was irreversible (stage 3-5), CoQ10 may be one of the solution to improve the testicular function and spermatogenesis.
Herein the reference 2 study we also applied the same laboratory platform to assess the effect of Metformin supplementation on the testicular function and spermatogenesis in the high-fat & high-cholesterol diet induced obesity mice model. Moreover the published article in 2020 has been cited by other two comprehensive review articles.
- Francesco Lotti, Sara Marchiani, Giovanni Corona, Mario Maggi. Metabolic Syndrome and Reproduction. Review Int J Mol Sci. 2021 Feb 17;22(4):1988. doi: 10.3390/ijms22041988.
- Alexander O Shpakov. Improvement Effect of Metformin on Female and Male Reproduction in Endocrine Pathologies and Its Mechanisms. Review Pharmaceuticals (Basel). 2021 Jan 8;14(1):42. doi: 10.3390/ph14010042.
Even with the limitations in our study that reviewer point out the CASA parameters was applied to evaluate the semen quality by Ghanayem et al. in 2010, there were other published studies utilized the light microscopy to analyze the semen quality in 2015 and 2018. To determine the sperm motility, we used a specialized counting chamber with special depth (Marienfeld, London, UK) to count the motile and immotile sperm. Additionally, the researcher has been trained to be familiar with whole process of the semen analyses.
- Wen-jie Yan, Yang Mu, Nan Yu, Tai-lang Yi, Yi Zhang, Xiang-li Pang, Dan Cheng, Jing Yang. Protective effects of metformin on reproductive function in obese male rats induced by high-fat diet. J Assist Reprod Genet (2015) 32:1097–1104.DOI 10.1007/s10815-015-0506-2
- V U Nna, A B A Bakar, A Ahmad, M Mohamed. Down-regulation of steroidogenesis-related genes and its accompanying fertility decline in streptozotocin-induced diabetic male rats: ameliorative effect of metformin. Andrology. 2019 Jan;7(1):110-123. doi: 10.1111/andr.12567. Epub 2018 Dec 4
We make all-out effort to achieve the aims in this study and the document has revised by professional English editing. Thanks for all the suggestions and we got loads out of it.
